# Family caregiver perspectives on strengths and challenges in the care of pediatric injury patients at a tertiary referral hospital in Northern Tanzania

**Elizabeth M. Keating** [1]*, **Francis Sakita**[2,3], **Maddy Vonderohe**[4], **Getrude Nkini**[2], **Ismail Amiri**[2], **Kelly Loutzenheiser**[5], **Bryan Young**[1], **Sharla Rent**[6], **Catherine A. Staton**[7,8,9], **Blandina T. Mmbaga**[2,3,10‡], **Melissa H. Watt**[11‡]

1 Department of Pediatrics, Division of Pediatric Emergency Medicine, University of Utah, Salt Lake City, Utah, United States of America, 2 Kilimanjaro Christian Medical Centre, Moshi, Tanzania, 3 Kilimanjaro Christian Medical University College, Moshi, Tanzania, 4 Department of Pediatrics, University of Utah, Salt Lake City, Utah, United States of America, 5 College of Nursing, University of Utah, Salt Lake City, UT, United States of America, 6 Department of Pediatrics, Duke School of Medicine, Durham, North Carolina, United States of America, 7 Global Emergency Medicine Innovation and Implementation (GEMINI) Research Center, Duke University Medical Center, Durham, North Carolina, United States of America, 8 Department of Emergency Medicine, Duke University Medical Center, Durham, North Carolina, United States of America, 9 Duke Global Health Institute, Duke University, Durham, North Carolina, United States of America, 10 Kilimanjaro Clinical Research Institute, Moshi, Tanzania, 11 Department of Population Health Sciences, University of Utah, Salt Lake City, Utah, United States of America

‡ BTM and MHW are contributed equally as co-senior authors
* elizabeth.keating@hsc.utah.edu

**Data Availability Statement:** The data for this paper is covered under a data sharing agreement, and thus we are unable to openly share these data.

## Abstract

### Background

Pediatric injuries are a leading cause of morbidity and mortality in low- and middle-income countries (LMICs). It is important that injured children get quality care in order to improve their outcomes. Injured children are nearly always accompanied by family member caregivers invested in their outcome, and who will be responsible for their recovery and rehabilitation after discharge.

### Objective

The purpose of this study was to identify family member caregiver perspectives on strengths and challenges in pediatric injury care throughout hospitalization at a tertiary hospital in Northern Tanzania.

### Methods

This study was conducted at a zonal referral hospital in Northern Tanzania. Qualitative semi-structured in-depth interviews (IDIs) were conducted by trained interviewers who were fluent in English and Swahili in order to examine the strengths and challenges in pediatric injury care. IDIs were completed from November 2020 to October 2021 with 30 family member caregivers of admitted pediatric injured patients. De-identified transcripts were

For data access requests, please contact our non-author KCMC representative Gwamaka William at gwamakawilliam14@gmail.com.

**Funding:** EMK was supported in this work by the Fogarty International Center of the National Institutes of Health (grant number D43 TW009337) and the Eunice Kennedy Shriver National Institute of Child Health and Human Development (grant number K23 HD112548). The content is solely the responsibility of the authors and does not necessarily represent the official views of the National Institutes of Health. The funders had no role in study design, data collection and analysis, decision to publish, or preparation of the manuscript. There was no additional external funding received for this study.

**Competing interests:** The authors have declared that no competing interests exist.

**Abbreviations:** EMD, Emergency Medicine Department; ICU, Intensive Care Unit; IDI, In-depth interview; KCMC, Kilimanjaro Christian Medical Centre; LMICs, Low- and middle-income countries.

synthesized in memos and analyzed through a team-based, thematic approach informed by applied thematic analysis.

## Results

Strengths and challenges were identified throughout the hospital experience, including emergency medicine department (EMD) care, inpatient wards care, and discharge. Across the three phases, strengths were identified such as how quickly patients were evaluated and treated, professionalism and communication between healthcare providers, attentive nursing care, frequent re-evaluation of a patient's condition, and open discussion with caregivers about readiness for discharge. Challenges identified related to lack of communication with caregivers, perceived inability of caregivers to ask questions, healthcare providers speaking in English during rounds with lack of interpretation into the caregivers' preferred language, and being sent home without instructions for rehabilitation, ongoing care, or guidance for follow-up.

## Conclusion

Caregiver perspectives highlighted strengths and challenges throughout the hospital experience that could lead to interventions to improve the care of pediatric injury patients in Northern Tanzania. These interventions include prioritizing communication with caregivers about patient status and care plan, ensuring all direct communication is in the caregivers' preferred language, and standardizing instructions regarding discharge and follow-up.

## Introduction

In 2000, traumatic injuries accounted for approximately 9% of the world's deaths, with more than 90% of these mortalities occurring in LMIC's [1]. Pediatric injuries are a leading cause of morbidity and mortality in low- and middle-income countries (LMICs). Unintentional injuries kill approximately 830,000 children every year, and more than 95% of child injury deaths (both intentional and unintentional) occur in LMICs. For each of these deaths, however, there are even more patients whose injuries result in significant morbidity. The World Health Organization estimates that up to half of pediatric injury patients who present to a hospital will be left with a lifelong disability [2]. For example, in a systematic review evaluating the morbidity and mortality of pediatric traumatic brain injuries in LMICs from January 2000 to May 2020, 170,224 cases of pediatric traumatic brain injuries were reported in 32 LMICs, of which 24% of patients had a reduction in their normal mental or physical function [3].

Given the significant impact of pediatric injuries on the morbidity and mortality of children in LMICs, it is crucial to evaluate the quality of their care not only from a medical perspective, but also from the perspective of the child and the family member caregiver. Injured children are nearly always accompanied by caregivers invested in their outcome, and who will be responsible for their recovery and rehabilitation after discharge. Thus, given the time and emotional investment of the caregivers, there is a significant gap in knowledge about their perspectives on the care both in-hospital and at discharge of these pediatric injury patients.

Family member caregivers have unique insights into the care of their injured children, and including them in research involving this population could help with finding unique solutions to challenges in caring for them. Involving patients and their caregivers in health research

gives them the opportunity to have more control over their healthcare and can help researchers to better understand their needs [4]. This is the overall premise of a newer field of science called community-based participatory research. Although there have been studies on caregiver perceptions of barriers to care in other areas of pediatric healthcare [5–8], there has been no such study to-date evaluating the strengths and challenges of care throughout hospitalization from the perspective of the caregivers of pediatric injury patients. Further, little research exists exploring the strengths and challenges to proper communication regarding care in LMIC pediatric care settings.

The aim of this study was to identify family member caregiver perspectives on strengths and challenges in pediatric injury care throughout hospitalization at a tertiary zonal referral hospital in Northern Tanzania. Identifying these challenges present at the tertiary care level will allow for quality improvement interventions to maximize and sustain the impact of pediatric trauma care at all levels of care.

## Methods

### Study setting

The study was conducted at Kilimanjaro Christian Medical Centre (KCMC), a tertiary zonal referral hospital in the Kilimanjaro region of Northern Tanzania. KCMC serves as a referral site for children with injuries requiring advanced imaging, subspecialist assessment and treatment, and higher level of care. Children with injuries at KCMC are initially seen in the Emergency Medicine Department (EMD). Those that require admission are admitted to one of six wards: two surgical wards, two pediatric wards, a burn ward, or an eye ward.

### Study design

This is a qualitative study involving semi-structured in-depth interviews (IDIs). We aimed to understand the strengths and challenges to care of pediatric injury patients at a tertiary hospital in Northern Tanzania from the family member caregiver perspective in order to define areas for quality improvement in the care of pediatric injury patients. This study received ethical approval from the institutional review boards at the Tanzanian National Institute for Medical Research (NIMR/HQ/R.8a/Vol.IX/3475; approved 7/20/2020), KCMC (1252; approved 10/19/2020), and the University of Utah (IRB_00134560; approved 9/1/2020). Written consent was obtained from the study participants to participate in this study.

### Participants

We completed 30 semi-structured IDIs with family member caregivers of pediatric injury patients admitted to KCMC from the EMD. Caregivers were family members including mothers, fathers, or other relatives that were the primary caregivers for the child. Caregivers of patients who died during hospitalization were not excluded from the study. The caregiver was enrolled if they were fluent in Swahili or English and could provide informed consent. Each pediatric injury patient was enrolled in a pediatric injury registry on arrival to KCMC, and during registry completion caregivers were identified who would be good informants for an interview. Close to the time of discharge, or shortly after death in occasions where the patient died, a purposive sample of caregivers were recruited face-to-face by a research assistant and invited to participate in an IDI. The plan to recruit 30 caregivers was informed by team-based discussions on when thematic saturation was most likely to occur. Previous work with in-depth interview data found that thematic saturation was present after 12 interviews [9,10]. However, in order to adequately represent our varied pediatric injury population, we felt that a

larger sample size of 30 would be more likely to achieve saturation. We thus approached caregivers of children of varying ages, mechanisms of injury, and injury types. Attention was given to interviewing caregivers who would be good informants, in that when completing the registry they appeared to be comfortable and forthcoming speaking about their child's healthcare experience.

## Qualitative procedures

IDIs were conducted by trained interviewers who were fluent in English and Swahili. The interview guide was developed in order to understand strengths and challenges in pediatric injury care at KCMC. Family member participants were asked about three phases of their child's care: their care in the EMD, throughout hospitalization, and surrounding discharge. For the EMD and hospitalization phases, they were asked about the care their child received including explanation of their child's condition and treatment, what they liked and disliked about the care, and things they would recommend are done differently to treat children in the future. For the discharge phase, if the child survived they were asked what happened when their child was discharged from the ward including if the caregiver felt ready for discharge and received adequate instructions on at home and follow-up care. The interview guide was translated from English to Swahili by two native Swahili speakers and checked for accuracy by a team of native speakers in translation meetings. The guide was pilot tested by the research team, both through training the interviewers and through going through the guide question by question with a representative parent to ensure questions were clear and well understood.

IDIs were completed from November 2020 to October 2021. IDIs took approximately one hour and were conducted in a quiet, private location within KCMC. The IDI occurred just prior to or after discharge from KCMC, or shortly after death in occasions where the patient died. Caregiver participants were provided a copy of the consent form and it was read aloud to them by the research assistant. Participants provided written informed consent, or a thumbprint if unable to write, prior to completing the IDI. Participants received 10,000 Tanzanian shillings (approximately 4 US Dollars) for time and transportation costs. IDIs were audio recorded, and interviewers made field notes during the interviews to assist with transcription. Recordings were transcribed and translated to English by the two bilingual Tanzanian research assistants. Other authors did not have access to information that could identify individual participants during or after data collection.

## Research team and reflexivity

IDIs were conducted by authors GN (female) and IA (male), who were research assistants at the time of the study. The research assistants completed two weeks of training that included qualitative interview strategies, review of the study protocol, and learning the qualitative interview guide. The interviewers were bilingual in English and Swahili, experienced in qualitative interviewing, and were trained by the principle investigator. There was no relationship between the interviewers and the participants prior to the interviews, but the participants were informed about the goals of the interviewer and reasons for doing the research prior to the interviews.

## Data analysis

De-identified data were analyzed through a team-based, thematic approach informed by applied thematic analysis [11]. After multiple readings, transcripts were synthesized in memos summarizing the data to identify and organize inductive themes, and inform preliminary codebooks [12,13]. Investigator BY wrote the memos after training by EMK and MHW.

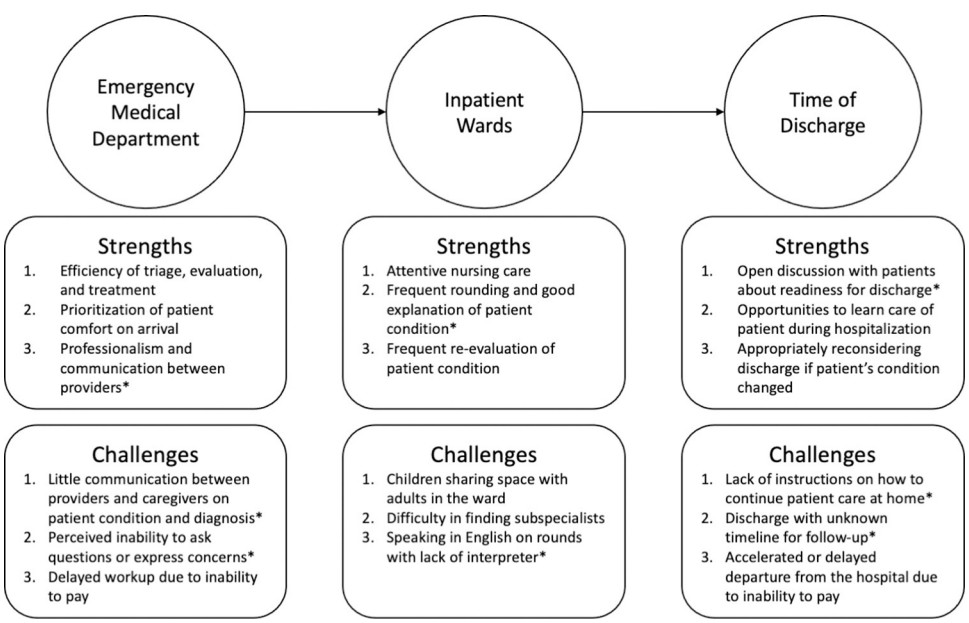

**Fig 1. Themes related to strengths and challenges by location phase.** *Strength or challenge related to communication.

Memos included representative quotes and were on average five single-spaced pages long. After reviewing the memos, emerging themes were reviewed by the study team, key constructs were defined, and edits were made to the codebook until team consensus was achieved. The memos were then coded in Dedoose© software, and the codebook was iteratively adapted to reflect new or emerging themes. All memos were double-coded by BY and SR, with intercoder discrepancy discussed by the research team and resolved by consensus, with EMK determining final coding if disagreements remained. As anticipated, we did reach saturation of themes after completion of all 30 IDIs, and thus decided it was not necessary to complete additional interviews.

## Results

The interviews revealed several strengths and challenges to care throughout hospitalization from the perspective of caregivers of pediatric injury patients (Fig 1). Three themes for both strengths and challenges at each point of hospitalization were coincidentally identified through analysis of the coded interviews.

### Emergency medicine department

**Strengths.** In reflecting on the strengths of the EMD experience, family member caregivers spoke about the efficiency of service and prioritization of their child's needs, and the professionalism of the healthcare provider team.

One strength identified by almost all caregivers interviewed was how quickly patients were seen, triaged, and treated. Multiple caregivers acknowledged that they felt as though their child was prioritized and evaluated with the appropriate urgency.

"I liked the way I arrived here and I got services on time. . .so generally everything went well because we didn't waste our time to wait for the providers."—Caregiver of a 3-year-old

male who presented to the EMD with a traumatic brain injury (diffuse brain injury), hemopneumothorax, and clavicle fracture secondary to a motor vehicle accident (IDI 14)

"I like the way they serve my child very fast: the way they dress her there, the way they check everything, and the way they gave her fluid because she doesn't have enough water in her body."—Caregiver of a 2-year-old girl who presented to the EMD with burn injuries (IDI 04)

Another strength identified is that the physicians were immediately attentive to the comfort of the patient, prioritizing pain control prior to beginning their testing and treatment.

"I liked the care which they established there. He got anti pain that made me happy that I didn't sit with him for long time while crying for pain. He got the first aid at emergency which helped to reduce his pain"—Caregiver of a 1-year-old male who presented to the EMD with a burn injury (IDI 23)

Caregivers also noted the professionalism of the providers in caring for the patients, as well as the apparent cooperation among the health care team.

"There are some things that I liked and I would still have them in mind about this hospital. The care really impressed me because my child was examined in high standard [of] professionalism. After few minutes the doctors would come in and check on my son and examine him. The child is given medication; the child is taken to do the tests. That kind of care is what satisfied me."—Caregiver of a 10-year-old child who presented to the EMD with a traumatic brain injury, cerebral edema, and an alveolar fracture secondary to a fall (IDI 02)

"They were so cooperative. The doctors communicated and gave each other instructions on what should be done to save [Patient]."- Caregiver of a 7-year-old female who presented to the EMD with a burn injury (IDI 26)

**Challenges.**   Reflecting on the challenges in the EMD, caregivers noted limited communication between the medical team and caregivers, the perceived inability to ask questions, and delays in care due to payment.

A challenge was the perceived limited communication between the medical team and the caregivers. This included lack of communication of the severity of the patients' injuries, the plan of evaluation (laboratory or radiographic), and the plan of care. Some caregivers described that staff would enter and exit the room, taking the patient for various testing without informing the family of the care plan. The status of the patient and severity of the injury was also not always understood by the caregivers, particularly in critical patients, and they had no concept of how badly the patient was injured.

"When they take blood, they didn't tell me that now we are taking his blood for further testing, although I overhead when they were talking to themselves that this child has this problem so we have to do a full blood picture. So, I heard that, but they didn't tell me anything when they came to take his blood. Also there is a thing–I think it was medicine–they only gave him through cannula but they didn't tell me what [the] medication [was] for."—Caregiver of a 4-year-old male who presented to the EMD with a humerus fracture (IDI 10)

"Nothing was explained to me about the condition of my son. They only consulted me when I was supposed to pay for the radiology."—Caregiver of a 9-year-old boy who

presented to the EMD with a traumatic brain injury (diffuse brain injury) secondary to a fall (IDI 15)

A second challenge was the perceived inability to ask questions to the EMD providers. This was largely assumed to be due to the healthcare providers being too busy caring for patients to answer caregiver questions.

"No, they told me nothing. At that very early stage it was not easy to ask [questions] as they were also very busy trying to save the life of the kids. From the [EMD] they did not tell me anything; they were so busy attending the babies they only asked what cause this severe burn and we answered them."—Caregiver of two children, a 3-year-old girl and an 11-year-old male who both presented to the EMD with burn injuries (IDI 07)

A final challenge identified by some caregivers were financial delays. Sometimes the work-up of the injured patient was delayed by failure to make initial payment.

". . .my child was soon reviewed by the doctors after arriving [to] the emergency but when we just mentioned we don't have money everything was stopped until when I sent that money that they needed."—Caregiver of a 13-year-old male who presented to the EMD with a traumatic brain injury (diffuse brain edema) due to near drowning, tension pneumo-thorax and pulmonary hemorrhage (IDI 09)

"I didn't like the way when I came in the EMD [I was] told to pay money before anything. For example, I paid 50,000 Tsh. I thought I would get all the treatment and come to pay my bill later, but here you pay for everything before. You have to pay for everything like tests, medicines. . . ."—Caregiver of a 6-year-old female who presented to the EMD with a burn injury (IDI 25)

## Inpatient wards

**Strengths.** Strengths during the hospital phase noted by caregivers included excellent nursing care, frequent rounding with appropriate explanations, and constant re-evaluation of their child's condition.

During hospitalization in the ward, a strength of the care received by the inpatient team highlighted by many caregivers was attentive nursing care. Multiple caregivers mentioned that the nursing staff was extremely attentive, and they were often more available than the doctors to answer caregiver questions about the patient's condition, care, and discharge plan.

"The nurses are so cooperative. They tell us everything they are doing and when you ask questions, they reply very well. I am so pleased and happy. . .They are so quick in giving the treatments. Even the time that she had to be transferred to ICU no time was wasted.

They quickly washed her wounds and brought her to the ICU."—Caregiver of a 7-year-old female who presented to the EMD with a burn injury (IDI 26)

Another strength noted was the clinical rounds, when providers explained the patients' condition, and caregivers felt free to ask questions.

"There are some things that I really liked. For instance, when the child is here in the ward, they do rounds in the morning, afternoon, and evening, and provide care for patients. You can also say anything or ask questions."—Caregiver of a 10-year-old child who presented to

the EMD with a traumatic brain injury, cerebral edema and an alveolar fracture secondary to a fall (IDI 02)

A final strength noted was that the patients' conditions were constantly re-evaluated and they were monitored closely, both by nursing staff and the physician team.

"I liked how they make follow-up of the patients; that they come to see them at morning, afternoon and evening. I mean it is very easy to notice if there is any change to the patient. Also if the patient is feeling pain, we are so free to check with providers so that he can get anti pain."—Caregiver of a 5-year-old male who presented to the EMD with a left eye injury (IDI 22)

**Challenges.**    Challenges that were described by the caregivers during the hospital stay included admitting children and adults to the same surgical ward, difficulty in locating subspecialists for care recommendations, and communication between the medical team and the caregivers.

The mixing of adult and pediatric patients in the surgical ward was seen as a weakness. Many felt as though it was damaging to their child's care, and inappropriate for them to see adult patients without clothes or hear them using foul language. Caregivers wanted their children to be hospitalized with other children in order to help their child relax and feel more comfortable in the hospital.

"Mostly I would recommend there should be separate wards for adults who are injured and the children. It's so sad to see how they are mixed up. . .these babies are being affected psychologically to see their elders naked, talking abusive language and all that. . . please think about them more."—Caregiver of a 7-year-old male who presented to the EMD with a femur fracture (IDI 19)

A second challenge noted by caregivers was that subspecialists were often very difficult to locate for further recommendations or medications, and it was often left to the caregivers to track them down.

"Burn structure is not well organized. You [are] supposed to get some of the medicines, but you are told to find the specialist to sign your forms of which this is not okay. The specialist should be available for the admitted patients not to suffer finding them."—Caregiver of two children, a 3-year-old girl and an 11-year-old male who both presented to the EMD with burn injuries (IDI 07)

A final challenge was that communication between the physicians and caregivers was not always in the preferred language of the caregiver, but rather the preferred language of the medical team, which was English. In addition, the ward rounds were in English. Thus, the caregiver was aware they were discussing their child, but did not know what they were saying. Further, often the patients themselves could understand more English than their caregivers. During ward rounds they would overhear the medical team discussing their condition in English, but not be able to fully understand or synthesize what they were hearing.

"Honestly, they have told me nothing about my child's condition. As I said before they normally communicate in a language that I don't really understand so it is hard for me to ask

them anything."—Caregiver of a 9-year-old boy who presented to the EMD with a traumatic brain injury (diffuse brain injury) secondary to a fall (IDI 15)

## Time of discharge

**Strengths.** In reflecting on the strengths at the time of discharge, caregivers noted that they felt involved in the decision of their child's readiness for discharge, they felt well prepared to care for their child at home given the education that nursing staff provided them throughout hospitalization, and their child was appropriately re-evaluated before discharge to ensure they were clinically stable to go home.

At time of discharge, multiple caregivers appreciated that they were generally allowed an opinion on the readiness of the patient to discharge home. When they did not feel ready, they felt that providers adequately addressed their concerns.

"Yes, I was free to ask and I asked them since I was not fine to go with the condition that [Patient] had. The doctor instructed me and told me no worries since you're still around will also take another X-ray that will show us the progress of [Patient]."—Caregiver of a 13-year-old male who presented to the EMD with a traumatic brain injury (diffuse brain edema), tension pneumothorax, and pulmonary hemorrhage due to near drowning (IDI 09)

A second strength noted was that due to the caregivers being at bedside with the patients at all times during hospitalization, they were able to learn how to care for their child from nursing staff and felt that they could mimic that care at the time of discharge.

"Medicines are the same and the time is the same as they used to give while we were here, so I have to proceed with the same medicine and the same hours."—A caregiver of a 4-year-old female who presented with an eyelid laceration (IDI 20)

A last strength during the discharge process was that the providers would appropriately re-evaluate the patient on the family's request and appropriately delay discharge if there was a change in their condition.

"When they removed the tube from her head then blood started to flow from her nose, so they said they will not let me go but they should wait to see her by that night then on Monday I will be allowed to go."—Caregiver of a 7-year-old female who presented with a traumatic brain injury (subdural and subarachnoid hemorrhages, depressed skull fracture), clavicle fracture and tibia fracture secondary to a motor vehicle accident (IDI 16)

**Challenges.** At the time of discharge, caregivers described challenges including lack of instructions for home care and follow-up appointments, and difficulty with affording the hospital bills.

Many caregivers shared that they were sent home without any instructions for rehabilitation or ongoing care of the patient's injury. Many caregivers reported feeling as though they did not receive sufficient education or preparation for the patient to continue to thrive and heal after discharge.

"I was not free [to ask questions about home care] since there was not anyone to ask. They only told me that you are now discharged. They didn't dispense me with medication, nor

any instructions. That happened during rounds, and they went to other patients."—Caregiver of a 5-year-old male who presented to the EMD with a humerus fracture (IDI 29)

A second challenge was that patients were often sent home without instructions for follow-up appointments for their injury.

"Yes, I needed to hear more information from them [about the follow-up appointment], but I did not get that chance."—Caregiver of a 9-year-old boy who presented to the EMD with a traumatic brain injury (diffuse brain injury) secondary to a fall (IDI 15)

A final challenge was either accelerated or delayed departure from the hospital due to a family's inability to pay the hospital bills. Some caregivers described feeling pressure to discharge earlier than they were medically comfortable so that their hospital bill would not be too high. Other caregivers described being medically ready for discharge but having to remain in the ward for a prolonged time until they could pay the hospital bills.

"Yes, I was not ready to go home. . .it is so expensive being there and we don't have enough money to cover the hospital bills. I was okay with the discharge though [Patient] was not well-cured. I had no option; raising money is difficult today while in the hospital."- Caregiver of a 9-year-old female who presented to the EMD with a femur fracture (IDI 24)

"I have stayed here for almost two weeks and all this is because of the hospital bills. I have no money. . .I am raising this child alone. . .the father does not give any support. I wonder where will I raise this huge amount of money from. . . I am just hoping the hospital will help me with the bill so we can go home."—Caregiver of a 10-year-old female who presented to the EMD with a burn injury (IDI 13)

## Discussion

It is imperative to understand challenges to pediatric injury care at a tertiary hospital in Northern Tanzania from the caregiver perspective. Recurring themes throughout the interviews included communication between the medical team and family member caregivers as both a strength and a challenge. The challenges identified by the caregivers in our study will be the foundation for future interventions including the introduction of a patient liaison who serves to update patient caregivers on their patient's status and the plan of care, ensuring all direct communication is in or translated to the caregivers' preferred language, and standardizing instructions regarding discharge and follow-up.

Many of the challenges mentioned by caregivers had to do with perceived inadequate communication between the healthcare providers and caregivers. Communication between patients, caregivers, and healthcare providers remains one of the largest challenges faced in pediatric healthcare. While quality communication practices are linked to improved patient outcomes [14], breakdowns in or lack of communication can lead to less than satisfactory healthcare experiences and a negative perception of patient and caregiver well-being [15]. Multicultural settings can introduce additional challenges to proper communication. The addition of language challenges and cultural differences can lead to disparities in treatment; decreased levels of trust between patients, caregivers, and providers; and a lack of culturally sensitive care [16,17]. These care challenges can result in patient and caregiver avoidant behavior when seeking care, errors in diagnosis and treatment, and suboptimal health education [18].

Caregivers in this study also consistently mentioned communication as a strength, expressing appreciation for detailed explanations of their patient's condition and open discussion about readiness for discharge. Research on patient-provider communication preferences in

sub-Saharan Africa is limited, and most studies on this topic are from high-income countries [19]. Preferred communication styles vary according to situation, cultural context and region, and patient characteristics [20,21]. The way providers communicate with patients can affect patient adherence and retention in outpatient chronic care in diabetes [22], human immunodeficiency virus treatment [19], and depression and chronic disease [23], as well as choice of first site for healthcare [19]. Research from rural Cameroon suggests that patients seek traditional healers in part because of the poor quality of patient-provider communication in the formal healthcare system [24]. Current literature from LMICs highlights caregiver communication as an important factor of treatment across all realms of pediatric care [25–27], but many cases of parental dissatisfaction in care and communication remain. Such research has led to the development and application of caregiver-focused care models in some LMICs [28], but one area in which investigation into parental perceptions and preferences is lacking is trauma care.

The need for emergent or urgent treatment in the trauma care setting in particular has the potential to exacerbate the communication challenges caregivers perceive in receiving care. Inadequate question-answering and delays in information-sharing can lead to caregivers feeling a lack of respect from the healthcare team, having unaddressed fears or concerns, and feeling excluded from their child's health [29]. The importance of prioritizing communication with caregivers cannot be understated, especially if the patient's status is critical. In environments such as the trauma care setting, a potential intervention to improve communication could be the introduction of a patient liaison who serves to update caregivers on their patient's status and plan of care.

Another challenge to communication, especially in the wards, was the lack of translation of patient rounds from English to Swahili, which created a language barrier that made it difficult for caregivers to understand their child's status and plan of care. This is a known challenge in many settings, and contributors to this problem include insufficient staffing, lack of healthcare provider time, and language proficiency [30]. Several studies have shown the importance of trained medical interpreters for ensuring effective communication between healthcare providers and patient families [31–33]. Language barriers in intercultural communication is a well-known challenge [34]. Many studies report about the positive effects of language concordance between the doctor and patient [35,36]. When providers are bilingual (as they often are in Swahili-speaking areas), it is especially important to speak to patient caregivers in the language in which they are most comfortable, in order to build rapport and break down communication hierarchies.

Many caregivers mentioned feeling inadequately prepared at discharge for ongoing care, rehabilitation, and follow-up of their pediatric patient. This brings up the topic of shared decision making, a concept that is beginning to be more recognized and discussed in hospitals in Sub-Saharan Africa [37]. One way to improve care would be to create more opportunities for patients and their caregivers to actively participate in decision making with healthcare providers. One study showed proof-of-concept evidence that more actively involving caregivers could significantly improve the quality of care in LMICs [38]. Regarding discharge instructions specifically, the content of pediatric hospital discharge instructions is highly variable, and even in high income countries many discharge instructions do not meet national standards. Often accessibility is limited by reading proficiency or discordant language of instructions [39]. Especially in the Tanzanian setting, standardized templates may be valuable to improve discharge instruction content and accessibility [39].

## Limitations

Our study did have limitations. First, there was a limited sample size with 30 caregivers interviewed. However, we had sufficient content to reach saturation in our themes. Further, since

all of the patients in this pediatric injury study were under 18 years old, these IDIs document the perspectives of their caregivers and not the patients' perspectives themselves. An additional limitation is that this study was done at a single tertiary referral hospital in Northern Tanzania, and extrapolating conclusions to other sites must be done with caution. While some findings may be site-specific, we expect that similar system-level issues exist in other tertiary hospitals in resource-denied settings. In addition, we only interviewed the caregivers of patients who had been admitted, which implies potentially more severe injuries. A future study could be done in which IDIs are conducted with caregivers of patients being discharged from the EMD to see if perspectives on EMD care change with patients with less severe injuries.

## Conclusions

This study highlighted strengths and challenges throughout the hospital experience from the family member caregiver perspective that will be the foundation for future interventions to improve the care of pediatric injury patients in Northern Tanzania. Even though this was a single site study, the methodology used and themes identified are very relevant to many settings in Sub-Saharan Africa and other LMICs [40]. Thus, these findings could possibly be extrapolated to improve the care of this patient population in other similar resource-limited settings. Our data suggests that potential interventions to improve family member caregiver experience include prioritizing communication with families about patient status and care plan, ensuring all direct communication is in or translated to the caregivers' preferred language, and standardizing discharge and follow-up instructions.

## Author Contributions

**Conceptualization:** Elizabeth M. Keating, Francis Sakita, Getrude Nkini, Ismail Amiri, Catherine A. Staton, Blandina T. Mmbaga, Melissa H. Watt.

**Data curation:** Elizabeth M. Keating, Getrude Nkini, Ismail Amiri, Bryan Young, Catherine A. Staton, Blandina T. Mmbaga.

**Formal analysis:** Elizabeth M. Keating, Maddy Vonderohe, Bryan Young, Sharla Rent.

**Funding acquisition:** Elizabeth M. Keating, Blandina T. Mmbaga, Melissa H. Watt.

**Investigation:** Getrude Nkini, Ismail Amiri, Melissa H. Watt.

**Methodology:** Elizabeth M. Keating, Maddy Vonderohe, Kelly Loutzenheiser, Bryan Young, Catherine A. Staton, Melissa H. Watt.

**Project administration:** Elizabeth M. Keating, Melissa H. Watt.

**Resources:** Elizabeth M. Keating, Francis Sakita, Getrude Nkini, Ismail Amiri, Sharla Rent, Catherine A. Staton, Blandina T. Mmbaga.

**Supervision:** Elizabeth M. Keating, Francis Sakita, Catherine A. Staton, Blandina T. Mmbaga, Melissa H. Watt.

**Visualization:** Elizabeth M. Keating, Maddy Vonderohe, Kelly Loutzenheiser, Sharla Rent, Catherine A. Staton, Blandina T. Mmbaga.

**Writing – original draft:** Elizabeth M. Keating, Maddy Vonderohe, Kelly Loutzenheiser.

**Writing – review & editing:** Elizabeth M. Keating, Francis Sakita, Getrude Nkini, Ismail Amiri, Bryan Young, Sharla Rent, Catherine A. Staton, Blandina T. Mmbaga, Melissa H. Watt.

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
