## [Decision Letter · Decision Letter 0]

30 Aug 2023

PONE-D-23-15434Caregiver perspectives on strengths and challenges in the care of pediatric injury patients at a tertiary referral hospital in Northern TanzaniaPLOS ONE

Dear Dr. Keating,

Thank you for submitting your manuscript to PLOS ONE. After careful consideration, we feel that it has merit but does not fully meet PLOS ONE’s publication criteria as it currently stands. Therefore, we invite you to submit a revised version of the manuscript that addresses the points raised during the review process. The manuscript was reviewed by three reviewers and their comments are incorporated below for your consideration.Reviewers's concerns that need more attention relate to the research methodology and discussion sections. There is also a bit of concerns about the related literature and you can enrich it.Please  consider the following:

Methodology: Clearly state how you chose/reached the sample size (e.g., how/when got reached saturation)           

                        Selection of participants, their impact ( e.g., selection bias from exclusion of those whose burnt patients died)

                       Elaborate more on the study procedure, including the piloting, etc.

Discussion: Elaborate more on the internal/external validity of the results. That is the clinical and/or socio-political implications in the study area and beyond

We look forward to receiving your revised manuscript.

Kind regards,

Janvier Hitayezu, M.D.

Guest Editor

PLOS ONE

Journal Requirements:

 "Fogarty International Center"

One or more authors are affiliated with the funder, but authors state that the funder had no role

Thank you for stating the following financial disclosure:

"EMK was supported in this work by the Fogarty International Center of the National Institutes of Health (D43 TW009337). The content is solely the responsibility of the authors and does not necessarily represent the official views of the National Institutes of Health. The funders had no role in study design, data collection and analysis, decision to publish, or preparation of the manuscript."

5. Please upload a new copy of Figure 1 as the detail is not clear. Please follow the link for more information: " ext-link-type="uri" xlink:type="simple">https://blogs.plos.org/plos/2019/06/looking-good-tips-for-creating-your-plos-figures-graphics/"
https://blogs.plos.org/plos/2019/06/looking-good-tips-for-creating-your-plos-figures-graphics/

Reviewers' comments:

Reviewer's Responses to Questions

**Comments to the Author**

1. Is the manuscript technically sound, and do the data support the conclusions?

Reviewer #1: No

Reviewer #2: Yes

Reviewer #3: Yes

2. Has the statistical analysis been performed appropriately and rigorously? 

Reviewer #1: No

Reviewer #2: Yes

Reviewer #3: Yes

3. Have the authors made all data underlying the findings in their manuscript fully available?

Reviewer #1: No

Reviewer #2: Yes

Reviewer #3: Yes

4. Is the manuscript presented in an intelligible fashion and written in standard English?

Reviewer #1: Yes

Reviewer #2: Yes

Reviewer #3: Yes

5. Review Comments to the Author

Reviewer #1: The study "Caregiver perspectives on strengths and challenges in the care of pediatric injury patients at a tertiary referral hospital in Northern Tanzania" aims to understand the experiences and views of caregivers who provide care to pediatric injury patients at a tertiary referral hospital in Northern Tanzania. The study explore the strengths and challenges faced by caregivers in providing care to these patients. The study could have had the possibility provide insights into the healthcare system and identify areas for improvement.

However, the study uses only qualitative methods to collect data from caregivers and limited sample size (i.e. 30 caregivers). The conclusion is erratic and wrote haphazardly. Too many ambiguous and unnecessary abbreviations were used. It does not provide any unique or intriguing aspect that would make readers eager to explore further. Additionally, the methodology or approach used in gathering the caregiver perspectives, which leaves the reader wondering about the reliability and validity of the data. Without this information, it becomes difficult to fully grasp the scope and depth of the study. The mention of a specific location, "Northern Tanzania," may limit the relevance and applicability of the findings to a wider audience. Readers outside of this region might view the study as having limited generalizability to their own settings, thereby diminishing its overall impact. Furthermore, It does not indicate how the findings could be utilized to improve the care of pediatric injury patients or address the challenges faced by caregivers.

In summary, the study lacks originality, fails to provide crucial information about the methodology, limited generalizability, and lacks emphasis on potential outcomes. These shortcomings diminish its effectiveness in capturing readers' interest and communicating the significance of the study.

Reviewer #2: I am pleased to provide a review of the intriguing paper titled "Caregiver perspectives on strengths and challenges in the care of pediatric injury patients at a tertiary referral hospital in Northern Tanzania." The entire paper is well-written; however, the revised version should address the following comments.

1. The introductory section appears to be insufficient in length. It is recommended to expand the content by incorporating additional paragraphs and introducing new references that are relevant to the study.

2. Please provide the list of abbreviations.

3.Please ensure that the reference number and date for your ethical approval are included.

4. Please provide more detailed explanations regarding the data availability statement.

Reviewer #3: In this qualitative study, the authors aimed to identify the strengths and challenges to care of pediatric injury patients at a tertiary hospital. The methodology and the results are well described. However the sample size is the major limitation of this study.

6. PLOS authors have the option to publish the peer review history of their article (what does this mean?). If published, this will include your full peer review and any attached files.

Reviewer #1: No

Reviewer #2: **Yes: **Sirwan Khalid Ahmed

Reviewer #3: No

---

## [Author Response · Author response to Decision Letter 0]

22 Sep 2023

Response to Reviewers: 

Journal Requirements:

Thank you for this instruction. We have edited our manuscript to meet the style requirements per the templates. 

 "Fogarty International Center"

One or more authors are affiliated with the funder, but authors state that the funder had no role

Thank you for stating the following financial disclosure:

"EMK was supported in this work by the Fogarty International Center of the National Institutes of Health (D43 TW009337). The content is solely the responsibility of the authors and does not necessarily represent the official views of the National Institutes of Health. The funders had no role in study design, data collection and analysis, decision to publish, or preparation of the manuscript."

Thank you for these instructions. The amended statement that declares all the funding or sources of support is as follows and has been edited as such in the manuscript page 23: “EMK was supported in this work by the Fogarty International Center of the National Institutes of Health (D43 TW009337) and the Eunice Kennedy Shriver National Institute of Child Health and Human Development (1K23HD112548-01). The content is solely the responsibility of the authors and does not necessarily represent the official views of the National Institutes of Health. The funders had no role in study design, data collection and analysis, decision to publish, or preparation of the manuscript. There was no additional external funding received for this study.” 

Thank you for these instructions. There are legal restrictions to sharing our data publicly as the data for this manuscript is covered under a data sharing agreement and is owned by KCMC in Tanzania. Here is our updated Data Availability statement: “The data for this manuscript is covered under a data sharing agreement, and thus we are unable to openly share these data. For data access requests, please contact our non-author KCMC representative Gwamaka William at gwamakawilliam14@gmail.com.”

Thank you for catching this error. We have amended the abstract on the online submission form so that it is identical to the abstract in the manuscript. 

5. Please upload a new copy of Figure 1 as the detail is not clear. Please follow the link for more information: https://blogs.plos.org/plos/2019/06/looking-good-tips-for-creating-your-plos-figures-graphics/" https://blogs.plos.org/plos/2019/06/looking-good-tips-for-creating-your-plos-figures-graphics/

Thank you for this instruction. A larger copy of Figure 1 has been uploaded that has clear detail.

Review Comments to the Author

Reviewer #1: The study "Caregiver perspectives on strengths and challenges in the care of pediatric injury patients at a tertiary referral hospital in Northern Tanzania" aims to understand the experiences and views of caregivers who provide care to pediatric injury patients at a tertiary referral hospital in Northern Tanzania. The study explore the strengths and challenges faced by caregivers in providing care to these patients. The study could have had the possibility provide insights into the healthcare system and identify areas for improvement.

Thank you for this summary. However, we believe this reviewer may have misunderstood the population that we interviewed, as we interviewed family member caregivers of pediatric injury patients, not the medical caregivers who provide care to the patients in the hospital. In order to make this more clear, we have added “family member” to caregivers in the abstract, introduction, methods, and conclusions. We have also revised the title to “Family caregiver perspectives on strengths and challenges in the care of pediatric injury patients at a tertiary referral hospital in Northern Tanzania.” 

However, the study uses only qualitative methods to collect data from caregivers and limited sample size (i.e. 30 caregivers). 

Thank you for noting this. Yes, this was a study using only qualitative methods. Regarding our sample size, we have provided more information regarding why this sample size was chosen on pages 7, lines 268-274: “The plan to recruit 30 caregivers was informed by team-based discussions on when thematic saturation was most likely to occur. Previous work with in-depth interview data found that thematic saturation was present after 12 interviews. However, in order to adequately represent our varied pediatric injury population, we felt that a larger sample size of 30 would be more likely to achieve saturation. We thus approached caregivers of children of varying ages, mechanisms of injury, and injury types. Attention was given to interviewing caregivers who would be good informants.” In addition, we did reach saturation as anticipated, which has been added to the end of the data analysis section in the methods in line 331.

The conclusion is erratic and wrote haphazardly. 

Thank you for this feedback. We have edited the conclusion so that it is more clear and focused.

Too many ambiguous and unnecessary abbreviations were used. It does not provide any unique or intriguing aspect that would make readers eager to explore further. 

Thank you for these comments. We have gone through and removed abbreviations that were unnecessary such as WHO, PI, TBI, HICs, HIV, SSA. We have spelled these out instead. We are now left with only five abbreviations that are used frequently throughout the manuscript. 

Additionally, the methodology or approach used in gathering the caregiver perspectives, which leaves the reader wondering about the reliability and validity of the data. Without this information, it becomes difficult to fully grasp the scope and depth of the study. 

Thank you for this comment that our methodology needs to be strengthened. We have strengthed our methodology by clearly stating how we chose the sample size (lines 268-274), how we selected participants (lines 272-274), and elaborated more on the study procedure including the piloting (lines 289-292). These methodology clarifications allow the reader to adequately assess the study’s internal validity. 

The mention of a specific location, "Northern Tanzania," may limit the relevance and applicability of the findings to a wider audience. Readers outside of this region might view the study as having limited generalizability to their own settings, thereby diminishing its overall impact. 

Thank you for this point. It is true that doing a study in one specific location may limit the relevence and applicability of the findings to a wider audience; however, we did state that this is a single center study and thus conclusions must be interpreted as such. While some of our study’s fundings may be site-specific, many systems issues will have relevance to other settings. This has been made clearer by adding the following to the Limitations section, lines 686-690: “An additional limitation is that this study was done at a single tertiary referral hospital in Northern Tanzania, and extrapolating conclusions to other sites must be done with caution. While some findings may be site-specific, we expect that similar system-level issues exist in other tertiary hospitals in resource-denied settings.” 

Furthermore, It does not indicate how the findings could be utilized to improve the care of pediatric injury patients or address the challenges faced by caregivers.

Thank you for this comment. In the first paragraph of our discussion, we give three examples of how the challenges identified in our findings could be addressed in lines 587-591: “The challenges identified by the caregivers in our study will be the foundation for future interventions including the introduction of a patient liaison who serves to update patient caregivers on their patient’s status and the plan of care, ensuring all direct communication is in or translated to the caregivers’ preferred language, and standardizing instructions regarding discharge and follow-up.” We hope that the additional changes made, incorporating the reviewers’ thoughtful feedback, help make the points made in this section of the discussion more clear. 

In summary, the study lacks originality, fails to provide crucial information about the methodology, limited generalizability, and lacks emphasis on potential outcomes. These shortcomings diminish its effectiveness in capturing readers' interest and communicating the significance of the study.

Thank you for this summary of your review. These peer-reviewed data will be the foundation for future interventions, which has been added to the Discussion line 588 and Conclusion line 697. Even though this was a single site study, the methodology used and themes identified are very relevant to many settings in Sub-Saharan Africa and other LMICs in general, which has been added to the Conclusion lines 698-700 along with a citation that demonstrates this (reference 40). We hope that by addressing your comments above we have made this study more effective in capturing readers’ interest and communicating the significance of the study.

Reviewer #2: I am pleased to provide a review of the intriguing paper titled "Caregiver perspectives on strengths and challenges in the care of pediatric injury patients at a tertiary referral hospital in Northern Tanzania." The entire paper is well-written; however, the revised version should address the following comments.

Thank you for this compliment. 

1. The introductory section appears to be insufficient in length. It is recommended to expand the content by incorporating additional paragraphs and introducing new references that are relevant to the study.

Thank you for this feedback. We have expanded the content of the introduction as suggested by adding an additional paragraph and introducing new references relevant to the study on page 5, lines 219-229. 

2. Please provide the list of abbreviations.

Thank you for this instruction. We have provided the list of abbreviations after the title page and before the abstract. 

3.Please ensure that the reference number and date for your ethical approval are included.

Thank you for catching this omission. We have added reference numbers and dates for all three ethical approvals in the methods section lines 262-263. 

4. Please provide more detailed explanations regarding the data availability statement.

Thank you for this request. There are legal restrictions to sharing our data publicly as the data for this manuscript is covered under a data sharing agreement and owned by KCMC in Tanzania. Here is our updated Data Availability statement: “The data for this manuscript is covered under a data sharing agreement, and thus we are unable to openly share these data. For data access requests, please contact our non-author KCMC representative Gwamaka William at gwamakawilliam14@gmail.com.”

Reviewer #3: In this qualitative study, the authors aimed to identify the strengths and challenges to care of pediatric injury patients at a tertiary hospital. The methodology and the results are well described. However the sample size is the major limitation of this study.

Thank you for this summary. Regarding our sample size, we have provided more information regarding why this sample size was chosen on pages 6-7, lines 268-274: “The plan to recruit 30 caregivers was informed by team-based discussions on when thematic saturation was most likely to occur. Previous work with in-depth interview data found that thematic saturation was present after 12 interviews. However, in order to adequately represent our varied pediatric injury population, we felt that a larger sample size of 30 would be more likely to achieve saturation. We thus approached caregivers of children of varying ages, mechanisms of injury, and injury types. Attention was given to interviewing caregivers who would be good informants.” In addition, we did reach saturation as anticipated, which has been added to the end of the data analysis section in the methods in line 331.

---

## [Editor Report · Decision Letter 1]

23 Oct 2023

PONE-D-23-15434R1Family caregiver perspectives on strengths and challenges in the care of pediatric injury patients at a tertiary referral hospital in Northern TanzaniaPLOS ONE

Dear Dr. Keating,

Thank you for responding to reviewers comments and submitting your revised manuscript to PLOS ONE. After careful consideration, we have realized some concerns were overlooked and we still feel that the manuscript has merit but does not fully meet PLOS ONE’s publication criteria as it currently stands. Therefore, we invite you to submit a new revised version of the manuscript that addresses the points raised during the review process.

Please note that, at this stage, the manuscript was not returned back to reviewers for further comments.

However, we believe the study procedure and methodology require more detailed clarifications and we want you to consider the following:

Please specifically state when saturation was achieved. Your statement at the line 191 does not tell whether this was achieved at exactly the 30^th^ IDI or before and at which IDI.

Explain further your selection of participants. It is essential to elaborate more on your inclusion/exclusion criteria and especially the rationale to exclude caregivers whose patients died, the impact of this exclusion on your findings.

It is observed that you have had the same number (3) of themes for strengths as well as for challenges, at EMD, wards, and at discharge. Please explain how this has happened, whether it was a mere coincidence or a specific reason and planned ahead of time.

Lastly, read carefully your manuscript to make sure all your statements are correctly spelt. E.g., the statement: “Authors did not have access to information that could identify individual participants during or after data collection” [Lines 185-186] is in contradiction with the subsequent statement: “IDIs were conducted by authors GN and IA, who were research assistants at the time of the study” [Lines 189-190].

On the lines 336-337, you are neutrally talking about a patient but then assigned him/her a female gender. “…If the patient is feeling, …..she can get anti pain”.

Please submit your new revised manuscript by Dec 07 2023 11:59PM If you will need more time than this to complete your revisions, please reply to this message or contact the journal office at plosone@plos.org. Please include the following items when submitting your revised manuscript:

A rebuttal letter that responds to each point raised by the academic editor and reviewer(s). You should upload this letter as a separate file labeled 'Response to Reviewers'.A marked-up copy of your manuscript that highlights changes made to the original version. You should upload this as a separate file labeled 'Revised Manuscript with Track Changes'.An unmarked version of your revised paper without tracked changes. You should upload this as a separate file labeled 'Manuscript'.If applicable, we recommend that you deposit your laboratory protocols in protocols.io to enhance the reproducibility of your results. Protocols.io assigns your protocol its own identifier (DOI) so that it can be cited independently in the future. For instructions see: https://journals.plos.org/plosone/s/submission-guidelines#loc-laboratory-protocols. Additionally, PLOS ONE offers an option for publishing peer-reviewed Lab Protocol articles, which describe protocols hosted on protocols.io. Read more information on sharing protocols at https://plos.org/protocols?utm_medium=editorial-emailutm_source=authorlettersutm_campaign=protocols.

We look forward to receiving your revised manuscript.

Kind regards,

Janvier Hitayezu, M.D.

Guest Editor

PLOS ONE
---

## [Author Response · Author response to Decision Letter 1]

21 Nov 2023

1. Please specifically state when saturation was achieved. Your statement at the line 191 does not tell whether this was achieved at exactly the 30th IDI or before and at which IDI.

Thank you for this question. While the analysis was iterative, there was a timelag from the data collection to transcription and translation, and then to data analysis. As such, we completed the established target of 30 IDIs. In analyzing the data, we confirmed that data saturation was reached at 30, and therefore we decided it was not necessary to complete additional interviews. This was clarified in lines 209-210 of the revised manuscript.

2. Explain further your selection of participants. It is essential to elaborate more on your inclusion/exclusion criteria and especially the rationale to exclude caregivers whose patients died, the impact of this exclusion on your findings.

Thank you for this instruction. The characteristics of participants identified is described in lines 139-155. We have elaborated on selection of participants by including information that each pediatric injury patient was enrolled in a pediatric injury registry on arrival to KCMC, and during registry completion caregivers were identified who would be good informants for an interview (lines 143-145). In addition, text was added to lines 154-155 that attention was given to interviewing caregivers who would be good informants, in that when completing the registry they appeared to be comfortable and forthcoming speaking about their child’s healthcare experience. In this revision we also clarified that death of the child in the hospital was not an exclusion criteria from participating in the study, and text has been added to make this clear in lines 141-142, 144, 168, and 178-179. 

3. It is observed that you have had the same number (3) of themes for strengths as well as for challenges, at EMD, wards, and at discharge. Please explain how this has happened, whether it was a mere coincidence or a specific reason and planned ahead of time.

Thank you for this inquiry. The identification of 3 themes for strengths and challenges at all phases of hospitalization did happen by mere coincidence. In this revision we have identified it as coincidence in lines 216-218 to address it up front. 

4. Lastly, read carefully your manuscript to make sure all your statements are correctly spelt. E.g., the statement: “Authors did not have access to information that could identify individual participants during or after data collection” [Lines 185-186] is in contradiction with the subsequent statement: “IDIs were conducted by authors GN and IA, who were research assistants at the time of the study” [Lines 189-190].

Thank you for pointing out this contradiction. We have corrected this in the revised manuscript. We have added “Other authors…” to distinguish that authors other than the research assistants did not have access to information that could identify individual participants during or after data collection. The remainder of the manuscript was carefully read to ensure all our statements are correctly spelled, and small wording changes were made as necessary throughout.

5. On the lines 336-337, you are neutrally talking about a patient but then assigned him/her a female gender. “…If the patient is feeling, …..she can get anti pain”.

Thank you for pointing this out. The "she" was in-fact a typo and has been corrected to "he," but in this instance the discordance between the neutral and gendered pronouns was taken from a quote from a parent, who is referring to his male child who was a pediatric injury patient. The parent included both neutral and gendered pronouns in their quote and thus we kept it this way as to not alter their words.

---

## [Editor Report · Decision Letter 2]

27 Nov 2023

Family caregiver perspectives on strengths and challenges in the care of pediatric injury patients at a tertiary referral hospital in Northern Tanzania

PONE-D-23-15434R2

Dear Dr. Keating,

We’re pleased to inform you that your manuscript has been judged scientifically suitable for publication and will be formally accepted for publication once it meets all outstanding technical requirements.

Kind regards,

Janvier Hitayezu, M.D.

Guest Editor

PLOS ONE

---

## [Editor Report · Acceptance letter]

8 Dec 2023

PONE-D-23-15434R2 

Family caregiver perspectives on strengths and challenges in the care of pediatric injury patients at a tertiary referral hospital in Northern Tanzania 

Dear Dr. Keating:

I'm pleased to inform you that your manuscript has been deemed suitable for publication in PLOS ONE. Congratulations! Your manuscript is now with our production department. 

Kind regards, 

on behalf of

Dr. Janvier Hitayezu 

Guest Editor

PLOS ONE